# WonderFree: Enhancing 3D World Generation via Video Diffusion Prior with Multi-view Consistency

## Abstract

3D scene generation from a single image has gained significant attention due to its potential to create immersive virtual worlds. However, a key challenge in current 3D generation methods is the limited explorability, which cannot render high-quality images during larger maneuvers beyond the original viewpoint, particularly when attempting to move forward into unseen areas. To address this challenge, we propose WonderFree, a model that enables users to generate 3D worlds with enhanced freedom to explore from diverse angles and directions. Specifically, we decouple this challenge into two key subproblems: novel view quality, which addresses visual artifacts and floating issues in novel views, and cross-view consistency, which ensures spatial consistency across different viewpoints. To enhance rendering quality in novel views, we introduce WorldRestorer, a data-driven video restoration model designed to eliminate floaters and artifacts. In addition, a data collection pipeline is presented to automatically gather training data for WorldRestorer, ensuring it can handle scenes with varying styles needed for 3D scene generation. Furthermore, to improve cross-view consistency, we propose ConsistView, a multi-view joint restoration mechanism that simultaneously restores multiple perspectives while maintaining spatiotemporal coherence. Qualitative visualization results demonstrate that WonderFree not only enhances rendering quality across diverse viewpoints but also improves global coherence and consistency. These improvements are further confirmed by CLIP-based metrics and a user study showing a 77.20% preference for WonderFree over WonderWorld.

## 1 Introduction

Creating a 3D world from text descriptions or a single image and freely exploring it has long been a human dream. In recent years, many studies (Yu et al., 2024a; Ni et al., 2025) have made significant progress. However, current methods for generating 3D scenes still have obvious limitations in explorability, as they can often only render high-quality images from a few specific viewpoints, making it difficult for users to explore freely from any angle.

The first significant challenge for limited explorability lies in novel view synthesis. When users freely explore the generated 3D world, the system must ensure high-quality image generation from every perspective. However, existing 3D scene generation methods (Höllein et al., 2023; Pu et al., 2024) rely on limited-view images or panoramas to supervise the creation of 3D worlds, constraining abilities to maintain quality across significant viewpoint variations.

The second core challenge lies in ensuring multi-view consistency in the generated 3D world. For continuous 3D environments, maintaining consistency across different views in terms of geometric structure and visual appearance within overlapping regions is crucial. However, existing methods (Yu et al., 2024a; Ni et al., 2025; Yu et al., 2024b) often focus solely on optimizing single-view output quality, leading to inconsistent overlaps across different views.

To address these challenges, we introduce WonderFree, a 3D world generation model that allows users to explore more freely from various angles and directions. Specifically, WonderFree first generates a coarse 3D world, and then progressively refines it through iterative optimization. In each iteration, videos from novel viewpoints are rendered, which may be filled with artifacts. These

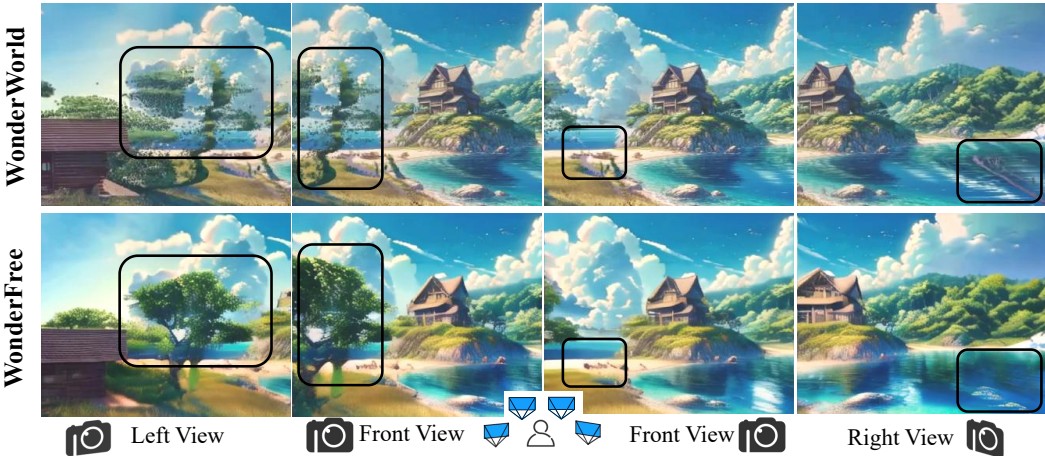

Figure 1: We compare our method with WonderWorld (Yu et al., 2024a) under novel views. The black-box regions exhibit noticeable artifacts in WonderWorld (Yu et al., 2024a), whereas Wonder-Free maintains clarity and visual consistency.

videos are then restored to obtain high-quality novel views, which are used to further refine the 3D world. To eliminate ghosting artifacts and distortions in these videos, we propose WorldRestorer, a restoration model fine-tuned from video generation frameworks (Blattmann et al., 2023). For training WorldRestorer, we design an automated data generation pipeline that constructs video restoration datasets across various scene styles. Meanwhile, to ensure consistency across different viewpoints, we integrate the ConsistView mechanism into WorldRestorer, a multi-view joint restoration approach, enhancing the spatiotemporal consistency and immersion of the generated 3D worlds.

As shown in Fig. 1, we present rendered images from multiple viewpoints after substantial translation. Existing state-of-the-art 3D generation methods, such as WonderWorld (Yu et al., 2024a), fail to render high-quality novel views, exhibiting noticeable floaters and structural distortions (e.g., deformation of trees). However, WonderFree robustly handles extensive viewpoint transformations, effectively maintaining spatial consistency across multiple perspectives.

The main contributions of this paper are summarized as follows:

- We introduce WonderFree, a framework designed to provide users greater flexibility when exploring 3D worlds from various angles and directions. This is achieved by designing an automated data generation pipeline and proposing WorldRestorer, a restoration model fine-tuned from a video generation framework to eliminate ghosting artifacts and distortions.

- For maintaining multi-view consistency, we integrate the ConsistView mechanism into WorldRestorer, which enhances spatial coherence across different viewpoints through a multi-view joint restoration approach.

- We conducted comprehensive experiments to demonstrate that WonderFree achieves state-of-the-art performance on CLIP-based metrics and user study preference rates, enhancing the quality of novel view synthesis and ensuring multi-view consistency.

## 2 RELATED WORK

### 2.1 3D WORLD GENERATION

The generation of 3D scenes from a single image has been explored by various methods. Some approaches first generate multiple views or panoramas of the scene and then convert them into 3D representations. For example, methods such as Text2Room (Höllein et al., 2023) and Lucid-Dreamer (Chung et al., 2023) start with a single input image and a user-provided textual description to generate multi-view images and construct a 3D world. On the other hand, methods like GenEx (Lu et al., 2024), Pano2Room (Pu et al., 2024), and DreamScene360 (Zhou et al., 2024a) use pre-trained

Figure 2: Overview of the WonderFree pipeline. WonderFree first builds a 3D world, then renders novel view videos along novel camera trajectories. WorldRestorer with the ConsistView is applied to restore these videos. The outputs are then used to refine the 3D world. This iterative process continues until the 3D world reaches a user-satisfactory level of visual quality.

text-to-panoramic diffusion models (Ye et al., 2025; Feng et al., 2023; Zhou et al., 2024b; Xiang et al., 2025) to synthesize coherent panoramas, which are then upscaled to create a 3D world that can be explored. Meanwhile, to enhance the interactivity of generated scenes, WonderWorld (Yu et al., 2024a) progressively constructs 3D worlds, allowing users to specify the content of the expanded 3D world through text. However, these methods typically use a limited number of views to supervise the generation of 3D worlds, resulting in the failure to render high-quality images when users attempt to move forward into unseen areas.

## 2.2 Scene Reconstruction with Diffusion Prior

To enhance the explorability and spatial consistency of 3D worlds, many works have attempted to leverage generative models (Blattmann et al., 2023; Rombach et al., 2022; Hong et al., 2022) to improve performance. WonderWorld (Yu et al., 2024a) introduces a single-view inpainting approach aimed at uncovering occluded regions in a generated scene using image inpainting techniques (Rombach et al., 2022). However, inpainting from a single view cannot ensure consistency across multiple viewpoints, and image inpainting-based methods struggle to maintain spatial consistency. Wonderland (Liang et al., 2024) adopts an approach based on the latent space of a video diffusion model (Blattmann et al., 2023) to construct a 3D reconstruction framework, which utilizes a feedforward mechanism to predict 3D Gaussian Splatting (3DGS) (Kerbl et al., 2023). Although this method improves spatial consistency, the generated 3D worlds are limited in scale. Meanwhile, some works (Ni et al., 2024; Zhao et al., 2025; Wang et al., 2025b; Zhao et al., 2024a; Wang et al., 2025a) in other fields have also attempted to use generative models to further enhance the spatial consistency of 3D worlds. However, these approaches are often restricted to one or two types of scenes, such as autonomous driving or indoor environments, and neglect consistency across multiple views. In contrast, WonderFree constructs a diverse video inpainting dataset through a dedicated pipeline, allowing WorldRestorer to handle ghosting artifacts and ensuring spatial consistency across multiple views with ConsistView.

## 3 Method

### 3.1 Overview of the WonderFree framework

Current 3D world generation methods often suffer from limited exploration capabilities due to supervision from only a limited number of views during scene generation. Therefore, WonderWorld (Yu et al., 2024a) adopts a single-view image inpainting approach to uncover occluded regions in a scene; however, it neglects spatial consistency. In contrast, WonderFree markedly enhances visual quality and ensures both spatial and temporal consistency across viewpoints.

Figure 3: The overview of the training WorldRestorer. An under-fitted Gaussian Splatting model is trained using ground truth videos to render degraded videos at different training stages, which are paired with corresponding GT videos for training WorldRestorer.

As shown in Fig. 2, WonderFree begins by generating a coarse 3D world from a single image using 3DGS. Then, an iterative optimization loop is applied to progressively refine the 3D world, involving two steps: (1) view restoration and (2) world refinement. During view restoration, we render videos from new trajectories in the coarse 3D world. Due to the absence of supervision at these novel viewpoints, the rendered videos often exhibit noticeable ghosting artifacts. Then, WorldRestorer, a diffusion-based restoration module, is applied to restore the corrupted novel views, conditioned on the rendered inputs. Crucially, to ensure spatial consistency, ConsistView is designed to jointly consider multiple viewpoints during the restoration process. In the world refinement stage, the restored videos serve as supervisory signals to improve the generated 3D world represented by 3DGS. This iterative loop continues until the generated 3D world attains high visual quality.

## 3.2 TRAINING AND INFERENCE OF WORLDRESTORER

Traditional 3D world generation methods (Yu et al., 2024a; Höllein et al., 2023; Chung et al., 2023; Ni et al., 2025) often suffer from visual artifacts when users explore novel views or trajectories that lack sufficient supervision. Although recent approaches (Yu et al., 2024a; Ni et al., 2025; Yu et al., 2024b) attempt to leverage powerful diffusion priors, most primarily rely on 2D priors and fail to effectively preserve spatial consistency across viewpoints. Additionally, video diffusion-based methods (Liang et al., 2024; Wallingford et al., 2024), while capturing temporal coherence, cannot iteratively refine the underlying 3D representation. To address these limitations, we introduce WorldRestorer, a diffusion-based restoration module specifically designed to repair degraded renderings from novel views, while effectively maintaining spatial coherence and structural fidelity. In the following sections, we detail the training and inference procedures of WorldRestorer.

**Training.** A major challenge in training WorldRestorer lies in the absence of datasets tailored for restoring artifacts arising from 3D world generation under novel trajectories. Therefore, we build a hybrid dataset of paired degraded and clean videos by first training an under-fitted Gaussian Splatting model using ground truth videos and rendering degraded videos at different training stages. Then we use this paired dataset to train WorldRestorer, as shown in Fig. 3. During training, we optimize the denoising network $\mathcal{D}$ using a conditional diffusion loss:

$$\mathcal{L}_{\mathcal{D}} = \mathbb{E}_{t,\eta} \left[ \|\eta - \mathcal{D}\left(\boldsymbol{z}_n, n, \boldsymbol{\mathcal{T}}\right)\|_2^2 \right], \tag{1}$$

where $\boldsymbol{z}_n$ is the latent representation corrupted by noise at step $n$, $\eta$ is the target noise to predict, and $\boldsymbol{\mathcal{T}}$ denotes the degraded input frames as the control condition.

**Inference.** After training WorldRestorer, we freeze its weights and deploy it to refine the synthesized 3D world. We first render videos from novel camera paths using the current world. Because the initial generation may be under-supervised, these renderings can exhibit artifacts. We pass them through the pretrained WorldRestorer to restore the frames. The restored videos are then fed back to further optimize the 3D world, enhancing cross-view consistency and visual quality. As illustrated in Fig. 4, WorldRestorer effectively suppresses ghosting in novel-trajectory renderings.

## 3.3 CONSISTVIEW

By constructing a video inpainting dataset and training a diffusion-based video generation model (Rombach et al., 2022; Blattmann et al., 2023; Wang et al., 2023b; Zhao et al., 2024b), we can

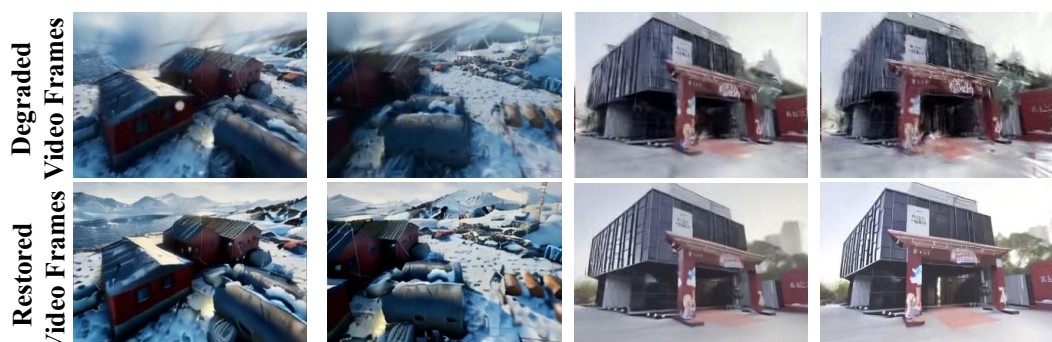

Figure 4: Examples of degraded video frames and their restored counterparts.

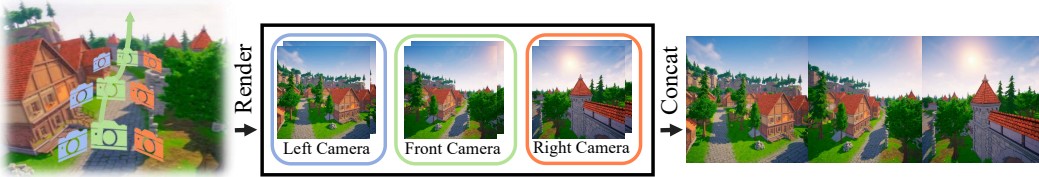

Figure 5: The overview of ConsistView. It first renders overlapping multi-view images, then concatenates them into a unified video for consistent restoration.

improve the exploration of 3D generated worlds. However, ensuring spatial consistency across multiple viewpoints remains a significant challenge. To mitigate this issue, existing methods (Yu et al., 2024d; Chen et al., 2025; Yu et al., 2024c) leverage reference images as conditional inputs to guide 3D world construction. While this improves alignment with target views, it remains inadequate for ensuring consistency across all generated viewpoints.

Therefore, we propose ConsistView within the WorldRestorer framework, with an overview illustrated in Fig. 5. Specifically, in the constructed 3D world, we move along a continuous path at fixed intervals and capture $K$ different viewpoint images at each location. These viewpoints are sampled with fixed angular offsets $\theta_k$ around the forward-facing direction, resulting in locally overlapping multi-view observations. The angular offsets are defined as:

$$\theta_k = (k - n)\Delta\theta, \quad k = 0, 1, \ldots, 2n \tag{2}$$

where $\Delta\theta$ represents the fixed angular step, and $n$ determines the total number of viewpoints on each side of the forward direction. Thus, $\theta_k$ spans the range from $-n\Delta\theta$ to $+n\Delta\theta$, corresponding to $2n + 1$ sampled viewpoints. The camera rotation at the $k$-th viewpoint can be defined as:

$$R_k = R_0 \cdot \text{Rot}_y(\theta_k) \tag{3}$$

$R_0$ is the rotation matrix for the forward view, and $\text{Rot}_y(\theta_k)$ represents a rotation around the vertical axis. At each time step $t$, a set of images is generated, forming a multi-view image sequence $\{x_t^k\}_{t=1, k=1}^{T, K}$, with each image having dimensions of $H \times W$. The images at the same time are then concatenated in a clockwise direction, resulting in a spatially unified video $\{x_t'\}_{t=1}^T$, with $x' \in \mathbb{R}^{3 \times H \times KW}$. This unified video $x'$ is then fed into WorldRestorer to obtain consistent restoration across all viewpoints. This approach is employed during both training and inference with $K = 3$ and an angular step $\Delta\theta$ of $60°$ to ensure robust consistency across views.

### 3.4 WORLDSCOPEDATASET: LARGE-SCALE AND MULTI-VIEW DATASET

To craft a model that restores corrupted novel views for 3D world generation, we propose a hybrid method combining real and synthetic data to create a diverse, multi-style, multi-view video dataset, named the WorldScopeDataset. The dataset primarily covers four categories: indoor environments (33%), urban landscapes (28%), natural terrains (21%), and stylized artistic scenes (18%). Based on the WorldScopeDataset, we construct a dedicated video restoration dataset by rendering degraded

| Epoch = 500 | Epoch = 1000 | Epoch = 1500 | Ground Truth |

Figure 6: Degraded frames rendered by 3DGS at different training epochs and their ground truth.

| Dataset | Year | Type | Source | #Images | #Scenes | Multi-view |
|---------|------|------|--------|---------|---------|------------|
| KITTI (Geiger et al., 2012a) | 2012 | U | Real | 15K | 1 | No |
| NYUv2 (Silberman et al., 2012) | 2012 | I | Real | 1.4K | 464 | No |
| SceneNN (Hua et al., 2016) | 2016 | I | Real | 502K | - | No |
| ScanNet (Dai et al., 2017a) | 2017 | I | Real | 2.5M | - | No |
| Replica (Straub et al., 2019) | 2019 | I | Real | 18 | - | No |
| nuScenes (Caesar et al., 2020) | 2020 | U | Real | 1.4M | 2 | No |
| Hypersim (Roberts et al., 2021) | 2021 | N | Synthetic | 77.4K | 461 | No |
| 3D-FRONT (Fu et al., 2021) | 2021 | N | Synthetic | 6,813 | - | No |
| ACID (Liu et al., 2021) | 2021 | N | Real | 2.1M | - | No |
| ScanNet++ (Yeshwanth et al., 2023) | 2023 | I | Real | 11.1M | 1006 | No |
| WorldScopeDataset | 2025 | I / U / N / A | Real & Synthetic | 23.4M | 6523 | Yes |

Table 1: Comparison with other 3D datasets. The types I, U, N, and A represent Indoor, Urban, Nature, and Artistic scenes.

videos from under-trained 3DGS, and pairing these renderings with their corresponding ground-truth frames. A detailed comparison with other mainstream 3D datasets is presented in Tab. 1.

For realistic scenes, we construct paired degraded and clean video sequences from real-world videos (Ling et al., 2024; Dai et al., 2017b; Sturm et al., 2012; Geiger et al., 2012b; Wang et al., 2021). Specifically, we begin by extracting a sequence of camera extrinsics $\mathcal{C} = \{C_l\}_{l=1}^{L}$ and sparse point clouds using COLMAP (Schonberger & Frahm, 2016), where $L$ denotes the total number of frames and $C_l$ represents the camera extrinsic parameters at frame $l$. To mimic the sparse-view supervision typically encountered in 3D world generation, we select frames at fixed intervals for 3DGS training. To capture varying reconstruction quality, we save $T$ model checkpoints at random intervals during training, resulting in under-trained models $\{\phi^{(t)}\}_{t=1}^{T}$. Each saved model $\phi^{(t)}$ is used to render a video along the original trajectory $\mathcal{C}$, producing a degraded video sequence:

$$\tilde{V}^{(t)} = \mathcal{R}\phi^{(t)}(\mathcal{C}), \tag{4}$$

where $\mathcal{R}\phi^{(k)}$ denotes the rendering function under model $\phi^{(k)}$. To simulate multiple viewpoints, each rendered video $\tilde{V}^{(k)}$ is divided into $N$ equal-length segments, with each segment representing a distinct virtual view along the trajectory. This strategy satisfies the training requirements of ConsistView to enable the model to learn spatial consistency across different perspectives. Due to early-stage underfitting, the rendered frames $\tilde{V}^{(k)}$ typically exhibit noticeable artifacts such as ghosting or floaters, as shown in Fig. 6. The corrupted videos are then paired with the original high-quality video $V_{\text{gt}}$ to form the dataset $D_{\text{real}}$:

$$\mathcal{D}_{\text{real}} = \left\{ \left( \tilde{V}^{(t)}, V_{\text{gt}} \right) \right\}_{t=1}^{T}. \tag{5}$$

For non-photorealistic scenes, we generate synthetic data using Unreal Engine (UE) (Engine, 2018), which provides a wide variety of customizable scene components. By composing these assets through automated scripts, we efficiently create a large number of diverse and stylized environments. In each scene, we define diverse continuous camera trajectories to simulate user exploration of the generated 3D world. At regular intervals along the trajectory, multiple viewpoints are simultaneously recorded with fixed angular offsets relative to the forward-facing direction, ensuring overlapping fields of view between adjacent perspectives (detailed in 3.3). Then, following the same data processing pipeline as for $\mathcal{S}_{\text{real}}$, we construct a new dataset denoted as $\mathcal{S}_{\text{synth}}$.

**WonderWorld** **WonderFree**

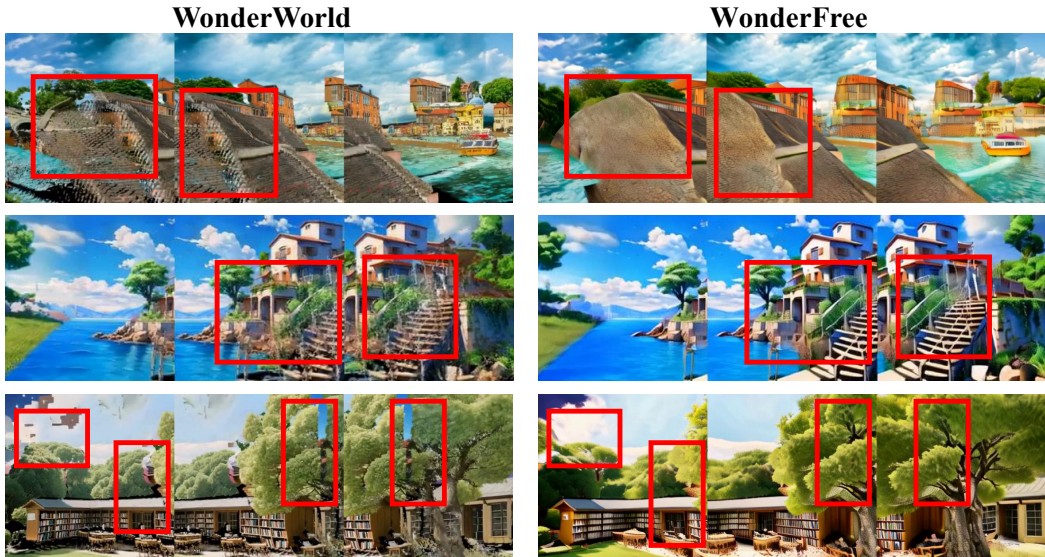

Figure 7: Qualitative comparisons between WonderFree and WonderWorld (Yu et al., 2024a) after large camera movements.

| Method | CLIP Score ↑ | CLIP Consistency ↑ | CLIP-IQA+ ↑ | Q-Align ↑ | CLIP Aesthetic ↑ |
|---|---|---|---|---|---|
| LucidDreamer (Chung et al., 2023) | 31.35 | 0.854 | 0.439 | 2.934 | 5.576 |
| Text2Room (Höllein et al., 2023) | 34.58 | 0.835 | 0.543 | 2.359 | 4.912 |
| DreamScene360 (Zhou et al., 2024a) | 30.24 | 0.765 | 0.426 | 2.145 | 4.873 |
| WonderJourney (Yu et al., 2024b) | 28.13 | 0.862 | 0.472 | 3.121 | 5.682 |
| WonderWorld (Yu et al., 2024a) | 32.28 | 0.913 | 0.560 | 3.437 | 6.123 |
| WonderTurbo (Ni et al., 2025) | 32.19 | 0.922 | 0.562 | 3.732 | 6.173 |
| WonderFree | **35.00** | **0.927** | **0.563** | **3.912** | **6.493** |

Table 2: Evaluation of novel view renderings for 3D world generation methods.

Finally, we merge $\mathcal{S}_{\text{synth}}$ and $\mathcal{S}_{\text{real}}$ to obtain $\mathcal{S}_{\text{all}}$. Training video restoration models on this multi-view dataset improves spatial consistency, making it well-suited for robust and coherent 3D scene generation.

## 4 EXPERIMENTS

In this section, we present our experimental setup, detailing the implementation procedures and the metrics used for evaluation. Subsequently, both quantitative and qualitative results are presented, highlighting the enhanced performance achieved by WonderFree. Lastly, we perform ablation studies to assess the individual contributions of each component.

### 4.1 EXPERIMENTAL SETUP

**Baselines.** We select various state-of-the-art 3D generation methods for comparison, including LucidDreamer (Chung et al., 2023) and Text2Room (Höllein et al., 2023), which generate 3D scenes via multi-view image synthesis; DreamScene360 (Zhou et al., 2024a), which creates panoramas and reconstructs them into full 3D environments3D world; and WonderJourney (Yu et al., 2024b), WonderWorld (Yu et al., 2024a), and WonderTurbo (Ni et al., 2025), which progressively generate 3D scenes based on user-provided text.

**Evaluation Metrics.** To evaluate the performance of 3D scene generation, we adopt the evaluation protocol introduced by WonderWorld (Yu et al., 2024a), which provides a framework for assessing generated scenes. This includes the use of several metrics such as the CLIP score, CLIP-based consistency, CLIP-IQA+, Q-Align, and the CLIP aesthetic score. These quantitative measures help

| Method | CLIP Score ↑ | CLIP Consistency ↑ | CLIP-IQA+ ↑ | Q-Align ↑ | CLIP Aesthetic ↑ |
|---|---|---|---|---|---|
| DIFIX3D+ (Wu et al., 2025) | 32.34 | 0.903 | 0.558 | 3.437 | 6.018 |
| ViewCrafter (Yu et al., 2024d) | 33.21 | 0.921 | 0.561 | 3.398 | 6.214 |
| MvSplat360 (Chen et al., 2025) | 32.76 | 0.912 | 0.559 | 3.413 | 6.134 |
| WonderFree | **35.00** | **0.927** | **0.563** | **3.912** | **6.493** |

Table 3: Quantitative comparisons of WonderFree with novel view synthesis methods.

| Method | Win Rate | Method | Win Rate |
|---|---|---|---|
| vs. LucidDreamer (Chung et al., 2023) | 94.10% | vs. Text2Room (Höllein et al., 2023) | 93.70% |
| vs. DreamScene360 (Zhou et al., 2024a) | 93.10% | vs. WonderJourney (Yu et al., 2024b) | 93.80% |
| vs. WonderWorld (Yu et al., 2024a) | 77.20% | vs. WonderTurbo (Ni et al., 2025) | 78.40% |

Table 4: Comparing the win rates of WonderFree in rendering novel views.

gauge the quality and coherence of the generated scenes from multiple perspectives. Additionally, we conduct a user study to gather subjective assessments of scene quality, ensuring a comprehensive evaluation that combines both objective metrics and human perception.

**Implementation Details.** To ensure an unbiased evaluation, we consider inputs from three baselines (LucidDreamer (Chung et al., 2023), WonderJourney (Yu et al., 2024b), and WonderWorld (Yu et al., 2024a)). For each of the 4 specified test cases, 8 unique scenes are created, resulting in a total of 32 generated scenes for assessment. To guarantee consistency across comparisons, we utilize the same camera configuration throughout scene generation and evaluation (details in the Appendix). Meanwhile, we select WonderWorld (Yu et al., 2024a) as our method to build a coarse 3D world.

## 4.2 MAIN RESULTS

**Quantitative Results.** As shown in Tab. 2, we compare WonderFree against several representative 3D scene generation approaches. WonderFree consistently achieves the best performance across all metrics by combining video diffusion priors with explicit multi-view consistency constraints. Compared with WonderWorld, it yields clear improvements of 13.82% on Q-Align and 6.04% on CLIP aesthetic, underscoring its advantages in both structural alignment and perceptual quality. In addition, our method substantially outperforms panorama-based baselines.

**Comparison with Novel View Synthesis Methods.** To demonstrate the necessity of proposing WorldRestorer for 3D world generation, we compare it with other methods that focus on novel view synthesis. We first construct a coarse 3D world and then apply these methods to refine it by reducing artifacts and filling in missing regions caused by occlusions. As shown in Tab. 3, WonderFree with WorldRestorer outperforms other methods. These methods (Yu et al., 2024d; Chen et al., 2024) typically focus on object-centric small scenes or purely image-based approaches, making them unsuitable for 3D world generation.

**User Study.** Additionally, we conducted a user study to evaluate the quality of 3D scenes generated by various methods. As shown in Tab. 4, the results indicate that WonderFree achieves win rates of 77.20% and 78.40% compared to WonderWorld and WonderTurbo, respectively.

**Qualitative Results.** As shown in Fig. 7, we conduct a qualitative comparison between Wonder-Free and WonderWorld under the same novel view. Notably, after significant camera movements, WonderWorld exhibits substantial distortions and blurriness in several regions, such as inside the red box. However, WonderFree significantly mitigates these issues, enhancing image quality and improving explorability. Additionally, WonderFree maintains consistency across multiple viewpoints and reduces missing content caused by occluded regions.

## 4.3 ABLATION STUDIES

**WorldRestorer.** As illustrated in Fig. 4, we present examples comparing degraded video frames with their restored counterparts. These examples clearly demonstrate that our WorldRestorer combined with ConsistView effectively removes visual artifacts and floaters while preserving visual

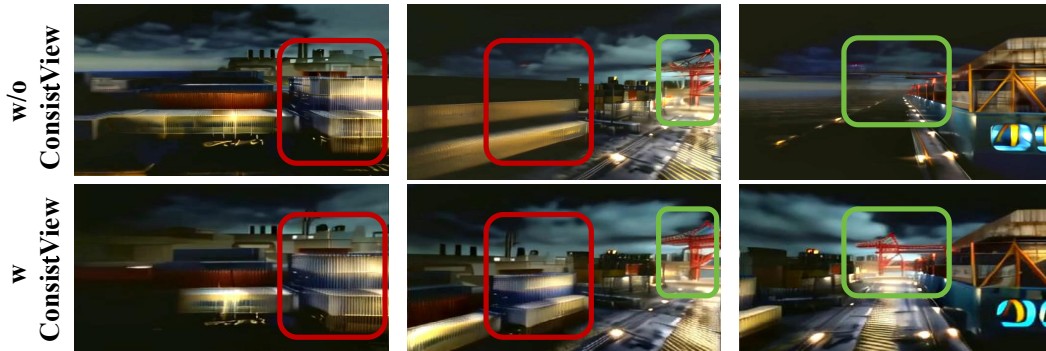

Figure 8: Visualization comparison of WorldRestorer with and without ConsistView. The first, second, and third columns correspond to the left, front, and right views, respectively. Overlapping regions between views are indicated by the red and green boxes.

| WorldRestorer | ConsistView | UE Data | CLIP Score ↑ | CLIP Consistency ↑ | CLIP-IQA+ ↑ | Q-Align ↑ | CLIP Aesthetic ↑ |
|---|---|---|---|---|---|---|---|
| | | | 32.28 | 0.913 | 0.560 | 3.437 | 6.123 |
| ✓ | | | 33.34 | 0.921 | 0.562 | 3.541 | 6.133 |
| ✓ | ✓ | | 34.42 | 0.926 | **0.563** | 3.829 | 6.481 |
| ✓ | ✓ | ✓ | **35.00** | **0.927** | 0.563 | **3.912** | **6.493** |

Table 5: Ablation study results of each module in WonderFree on novel view rendering.

coherence, including consistent lighting and shading effects. As shown in Tab. 5, WorldRestorer alone notably improves the baseline performance across various evaluation metrics.

**ConsistView.** As shown in Fig. 8, we compare WorldRestorer with and without ConsistView under challenging low-light conditions. Without ConsistView, the model restores individual views and removes floaters effectively, but fails to guarantee cross-view consistency, causing noticeable discrepancies across perspectives. In contrast, incorporating ConsistView enforces multi-view alignment, producing coherent and stable results across all views. Quantitative results in Tab. 5 confirm these findings, where the ConsistView-enhanced model achieves superior performance on all metrics. This highlights that while WorldRestorer is strong for single-view restoration, ConsistView is essential for robust multi-view consistency in 3D scene understanding.

**WorldScopeDataset.** We show in Tab. 5 that incorporating UE-collected data into WorldScope-Dataset leads to a significant improvement in CLIP aesthetic scores. This is because 3D world generation often requires handling inputs with diverse visual styles, whereas traditional 3D datasets tend to focus on a few specific domains, which limits their ability to capture such variety. In contrast, UE data, in the form of assets, introduces a broader range of visual diversity, enabling the model to better adapt to a variety of input styles and enhancing the aesthetic quality of the generated outputs.

## 5 CONCLUSION

Despite recent progress in 3D scene generation from a single image, current methods still struggle to render high-quality images during larger viewpoint maneuvers beyond the original perspective. To address this, we decouple this challenge into two key subproblems: novel view quality, which addresses visual artifacts in novel views, and cross-view consistency, which ensures spatial consistency across different viewpoints. Meanwhile, we propose WonderFree, a framework that integrates video diffusion priors and emphasizes cross-view consistency to mitigate these limitations. Specifically, to improve novel-view quality, we introduce WorldRestorer, a restoration model capable of eliminating visual artifacts and distortions. Additionally, to ensure cross-view consistency, we propose ConsistView, a multi-view joint restoration mechanism designed to enhance spatial consistency in the overlapping regions across different viewpoints. Moreover, we construct WorldScopeDataset, a dataset comprising large-scale, diverse scenes with multi-view data, for training WorldRestorer. Experimental results demonstrate that WonderFree significantly enhances rendering quality, both in CLIP-based metrics and user studies, thereby facilitating a more flexible exploration of 3D worlds.

## 6 REPRODUCIBILITY STATEMENT

Our method is described in detail in Section 3, which includes specific parameter settings and implementation details. We provide a comprehensive description of the datasets utilized, including their origins, characteristics, and any preprocessing steps undertaken to ensure data quality. Furthermore, we delve into the significance of choosing these particular datasets and discuss how they align with the objectives of our study. To ensure reproducibility, all relevant code and datasets will be made publicly available upon publication. This will allow other researchers to easily replicate our experiments, validate our findings, and build upon our work. We believe that transparency in research practices is crucial for advancing knowledge in our field and fostering collaboration among peers.

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

# A  IMPLEMENTATION DETAILS

**Please compare the two images below carefully.**
**Which one appears to be of better quality (with fewer errors) to you?**

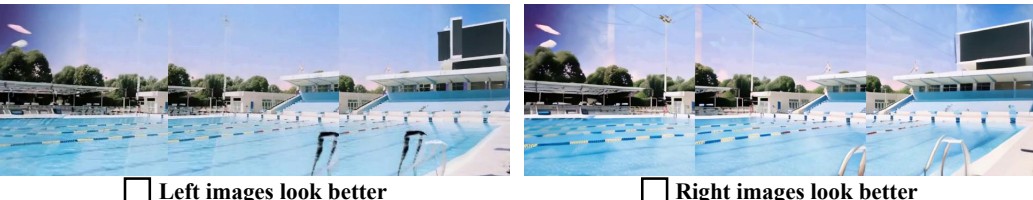

☐ **Left images look better**                    ☐ **Right images look better**

Figure 9: Screenshot of the user study survey interface.

**Metrics.** To assess the quality of 3D world generation, several metrics such as the CLIP score (Radford et al., 2021), CLIP consistency (Radford et al., 2021), CLIP-IQA+ (Wang et al., 2023a), Q-Align (Wu et al., 2023), and the CLIP aesthetic (Radford et al., 2021) score are utilized to evaluate both semantic alignment and visual aesthetics. Specifically, the CLIP score (Radford et al., 2021) measures semantic relevance by computing the cosine similarity between CLIP embeddings of the input textual scene descriptions and the rendered images. CLIP consistency (Radford et al., 2021) evaluates semantic coherence across various viewpoints by comparing their respective embeddings with the embedding of a central reference view. CLIP-IQA+ (Wang et al., 2023a) combines perceptual image quality assessment methods with deep learning techniques to deliver a comprehensive evaluation of overall image quality. Q-Align (Wu et al., 2023) assesses visual quality by training large multimodal models to predict human-aligned, discrete quality ratings, enhancing interpretability and generalization. Lastly, the CLIP aesthetic (Radford et al., 2021) score assesses the visual attractiveness of rendered scenes by considering critical aesthetic elements such as composition, contrast, and color harmony.

**Camera Trajectories.** In most 3D scene generation methods (Yu et al., 2024a; Höllein et al., 2023; Chung et al., 2023; Pu et al., 2024; Ni et al., 2025), simple panoramic camera paths are commonly used to evaluate the quality of generated worlds. However, these paths often do not sufficiently capture interactive movements typically performed by users during actual exploration. Therefore, we design more complex camera trajectories to simulate realistic exploration scenarios better. In addition to the conventional panoramic camera paths, we introduce five additional camera movements: camera moves forward, forward-left, forward-right, translates left, and translates right. During testing, we select views at regular intervals along each trajectory as novel views, and the final evaluation metrics are computed by averaging results across all paths.

**User Study.** We recruit 35 volunteers for a human preference evaluation and conduct the survey using Tencent Forms. The entire process is fully anonymous. Participants are presented with multi-view images generated by different algorithms, with the top-bottom order randomized, similar to the layout shown in Fig. 9. For each image pair, participants are asked to choose one of two options: "Top image looks better" or "Bottom image looks better." The instruction is: "Please compare the two images below carefully. Which one appears to be of better quality (with fewer errors) to you?"

# B  THE USE OF LARGE LANGUAGE MODELS

In this study, large language models (LLMs) are utilized exclusively for the purpose of textual enhancement. Their primary role is to refine the manuscript to enhance readability, clarity, and coherence, without affecting the methodology or the outcomes of the experiments. This approach guarantees that the function of LLMs is limited to language improvement, while all technical contributions and results are generated independently.

# C  QUALITATIVE RESULTS

As shown in Fig. 10 to Fig. 27, we present comparisons between WonderFree and WonderWorld (Yu et al., 2024a) under various styles and different trajectories, clearly demonstrating

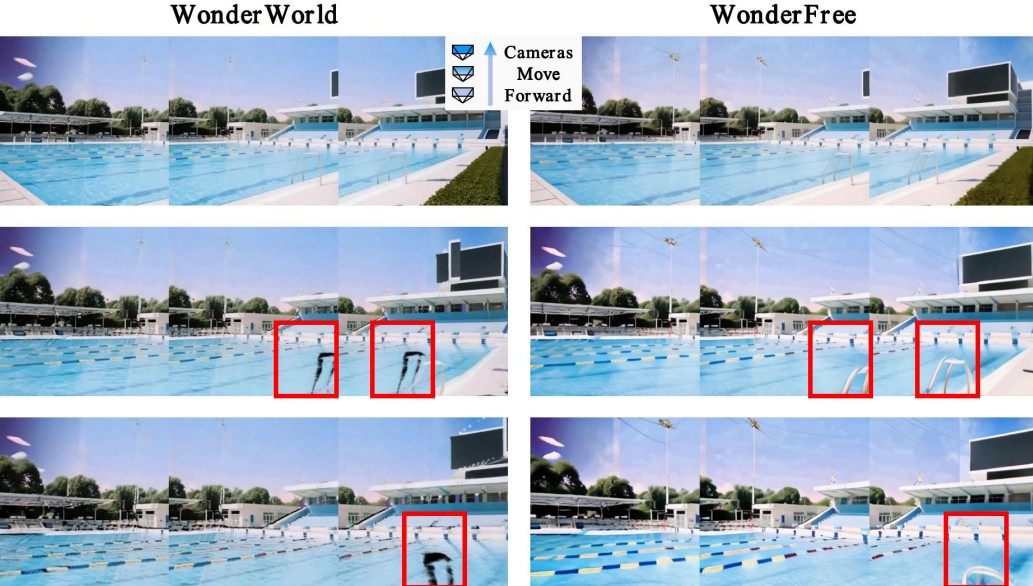

Figure 10: Qualitative examples.

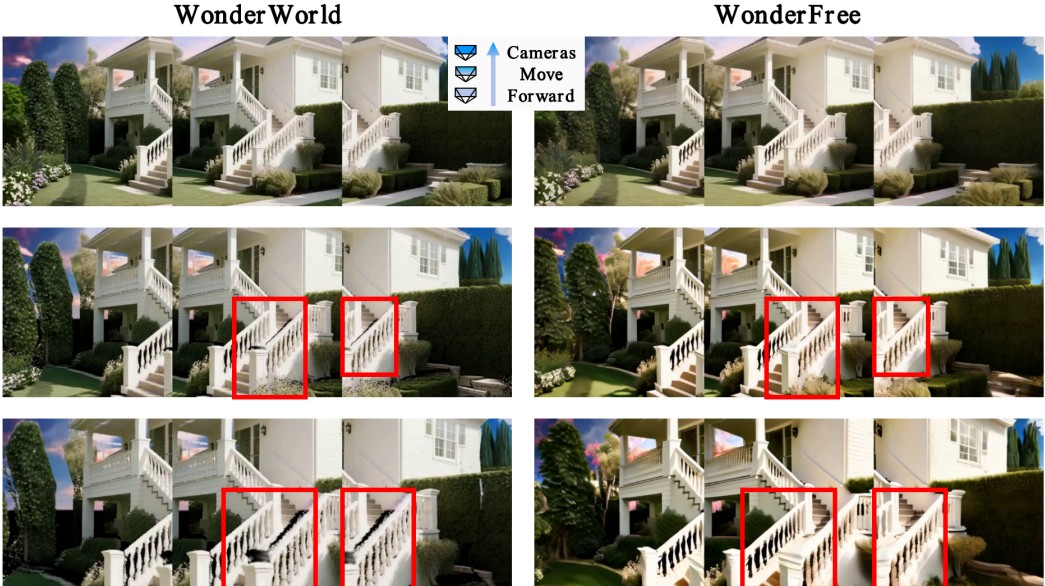

Figure 11: Qualitative examples.

the superiority of WonderFree. Meanwhile, we provide two videos, one comparing Wonder-World (Yu et al., 2024a) and WonderFree (`videos/comparison1.mp4`), and the other illustrating how WonderFree effectively refines the coarse 3D world compared to the original version (`videos/comparison2.mp4`).

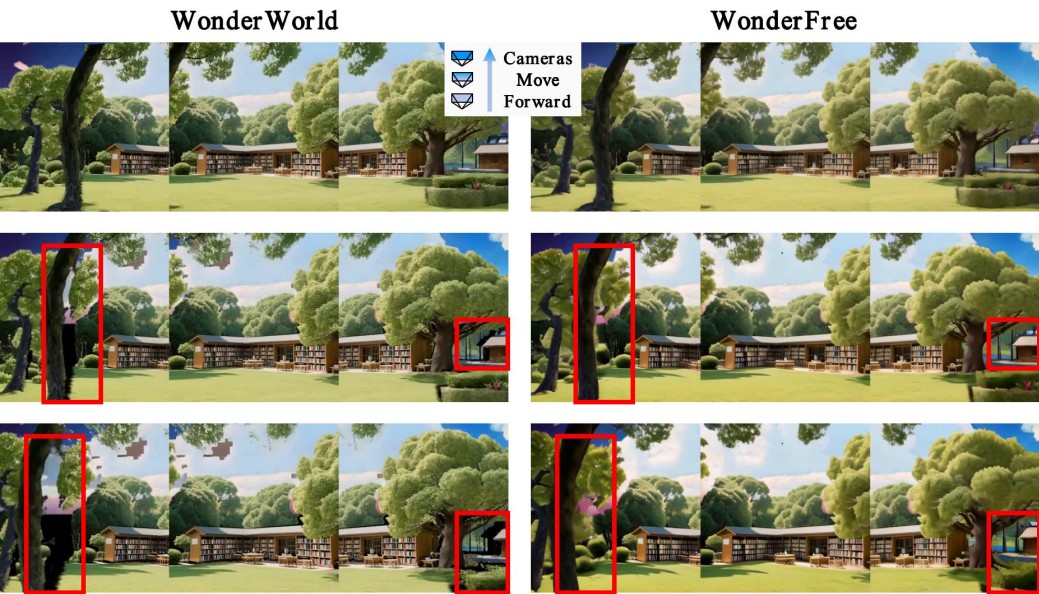

Figure 12: Qualitative examples.

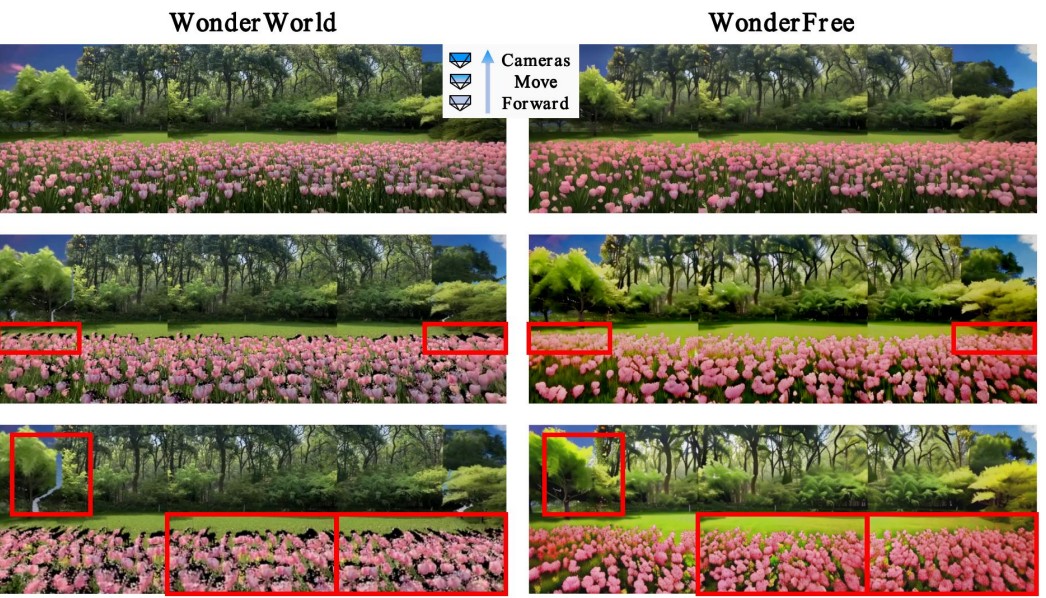

Figure 13: Qualitative examples.

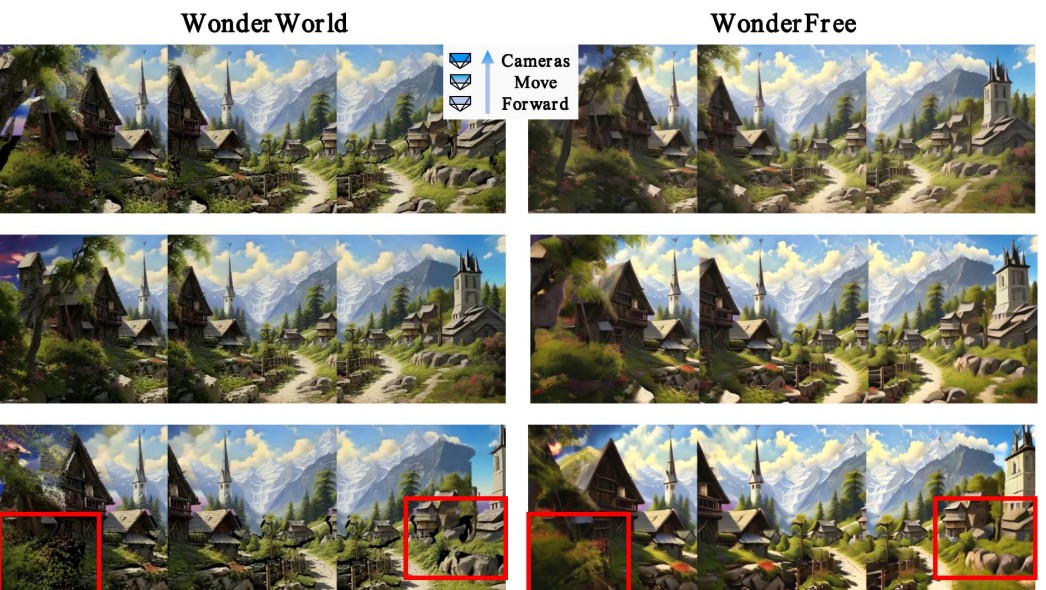

Figure 14: Qualitative examples.

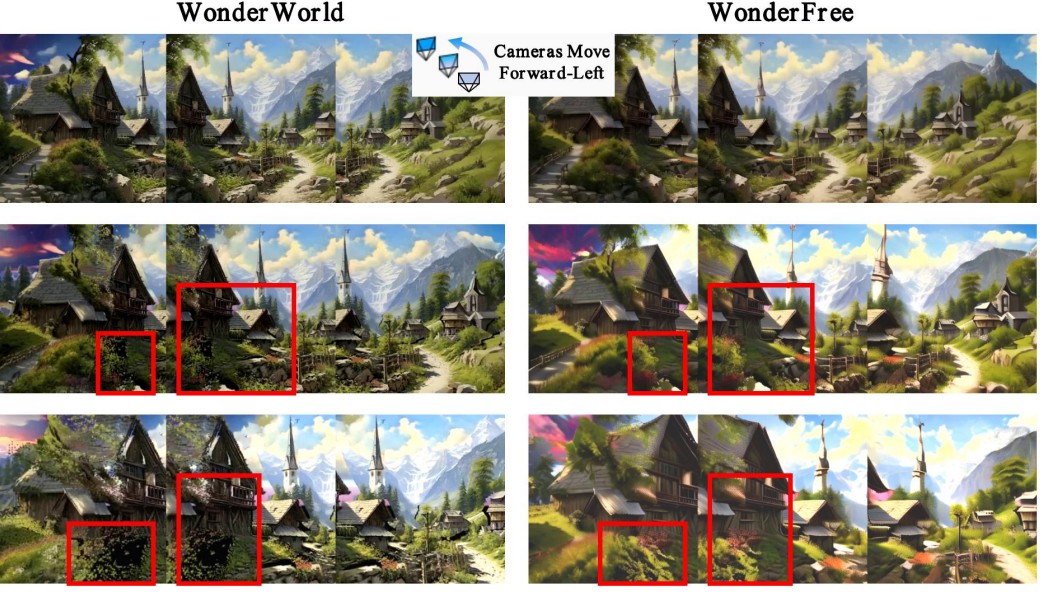

Figure 15: Qualitative examples.

WonderWorld                                      WonderFree

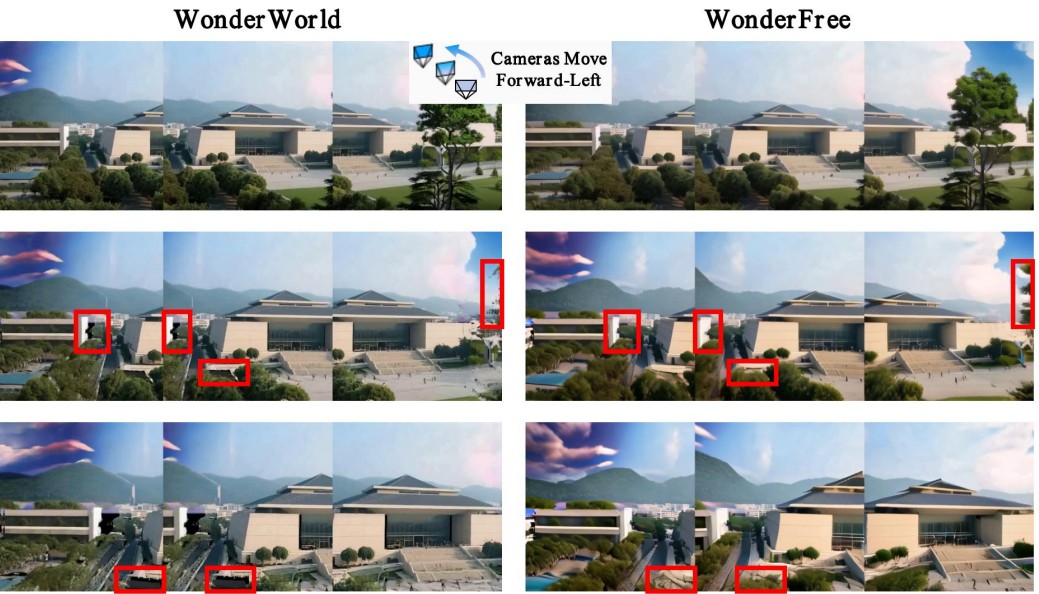

Figure 16: Qualitative examples.

WonderWorld                                      WonderFree

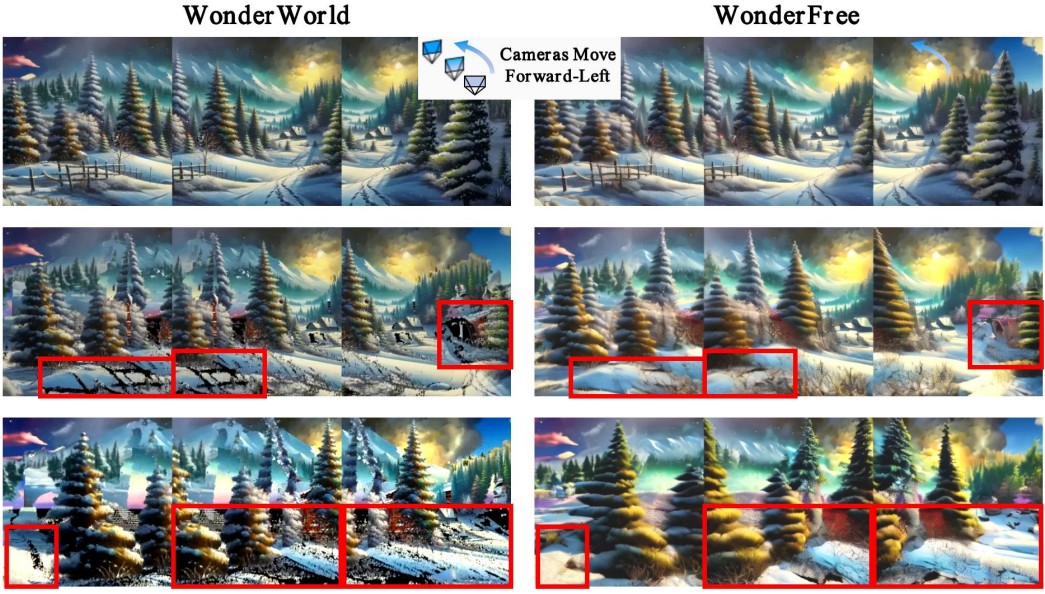

Figure 17: Qualitative examples.

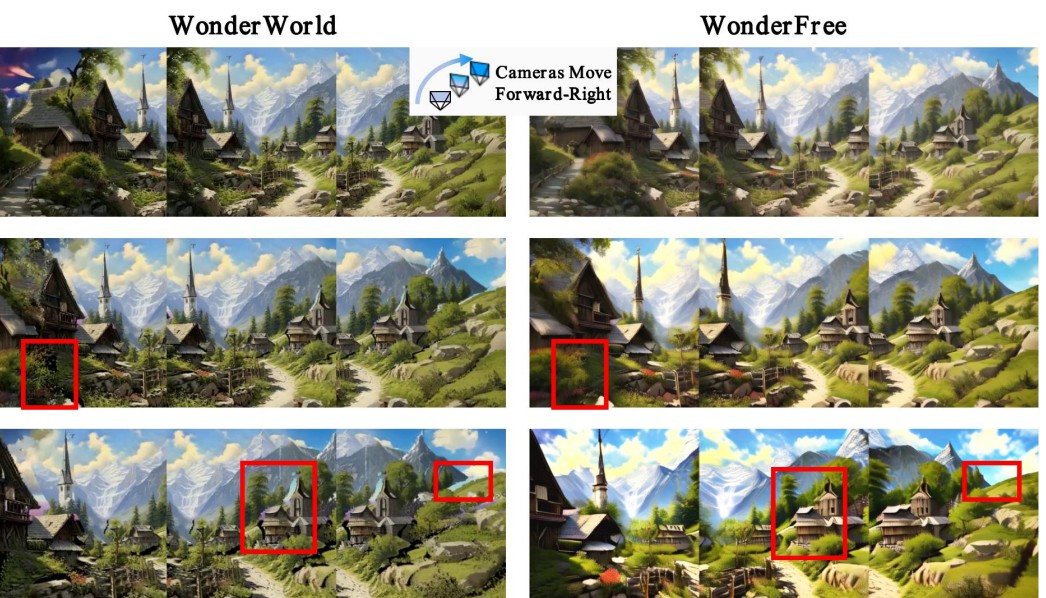

Figure 18: Qualitative examples.

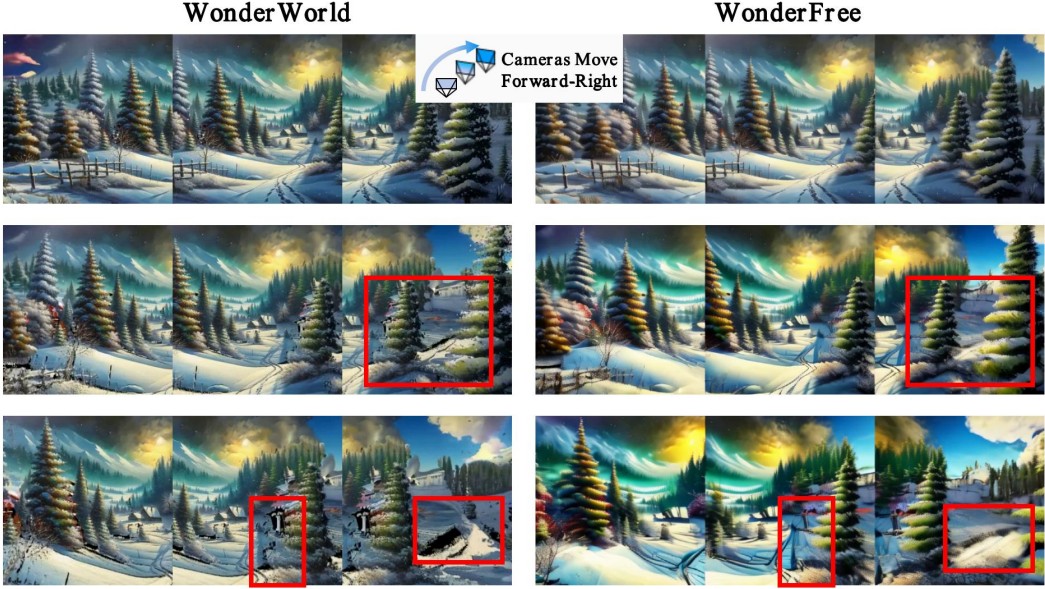

Figure 19: Qualitative examples.

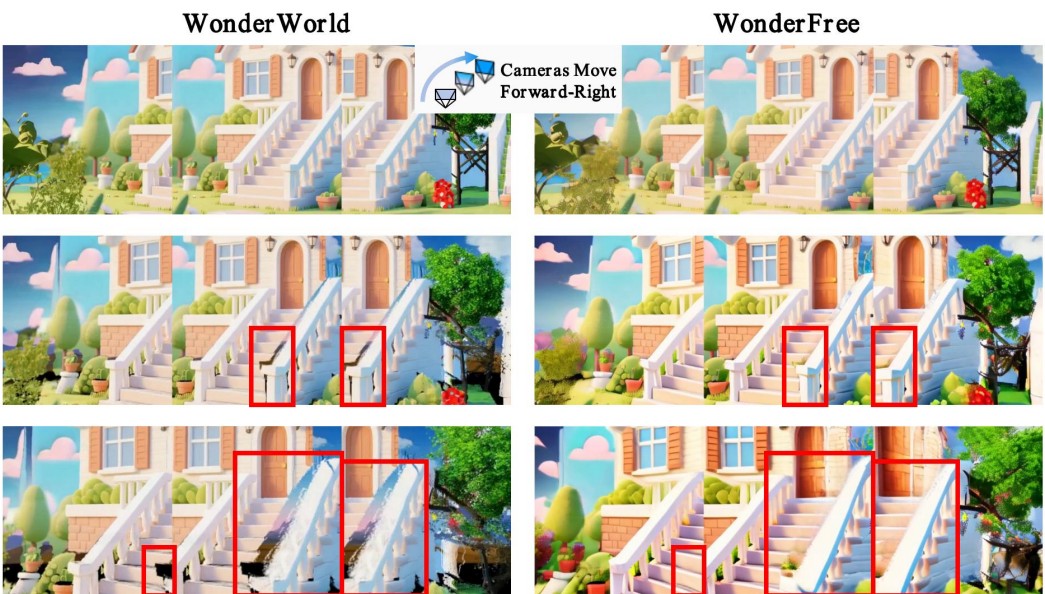

Figure 20: Qualitative examples.

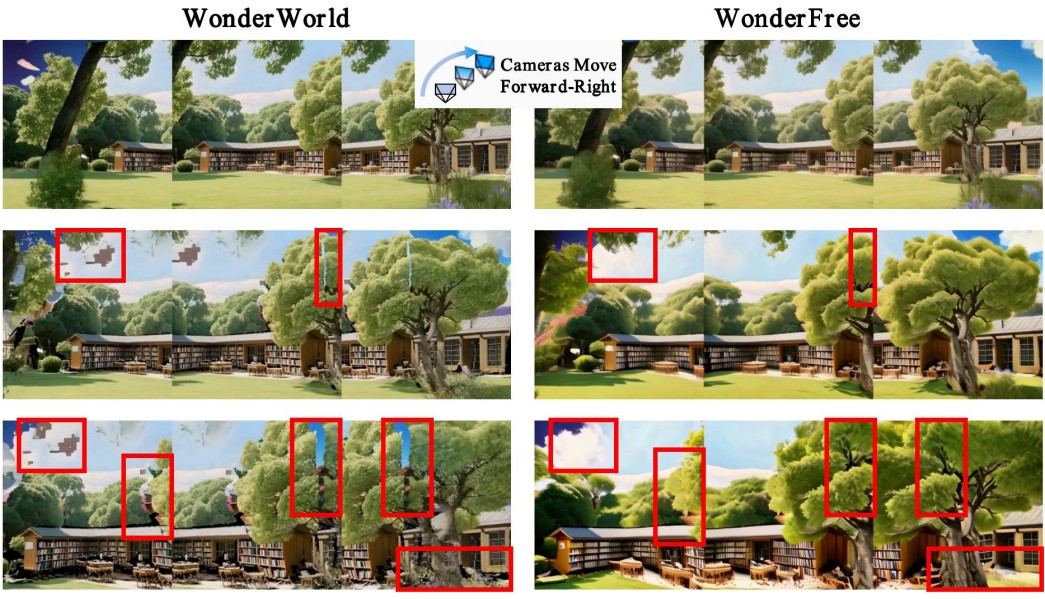

Figure 21: Qualitative examples.

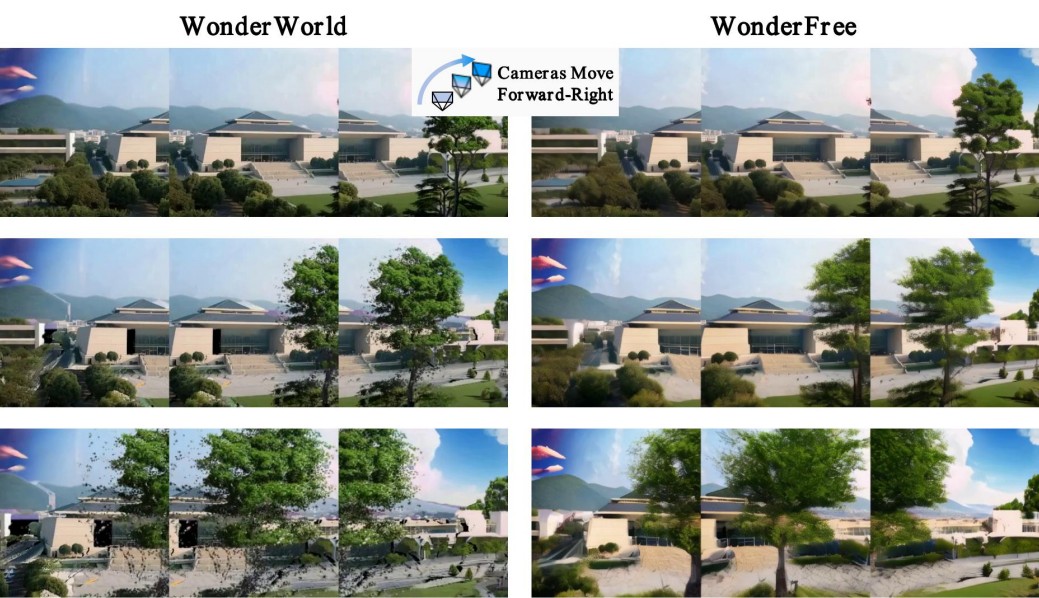

Figure 22: Qualitative examples.

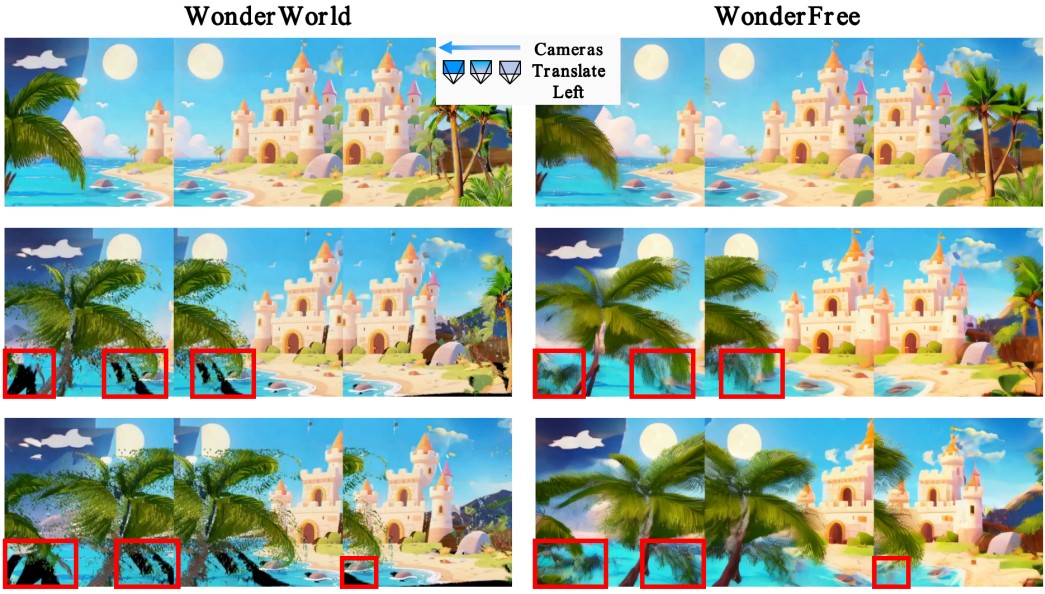

Figure 23: Qualitative examples.

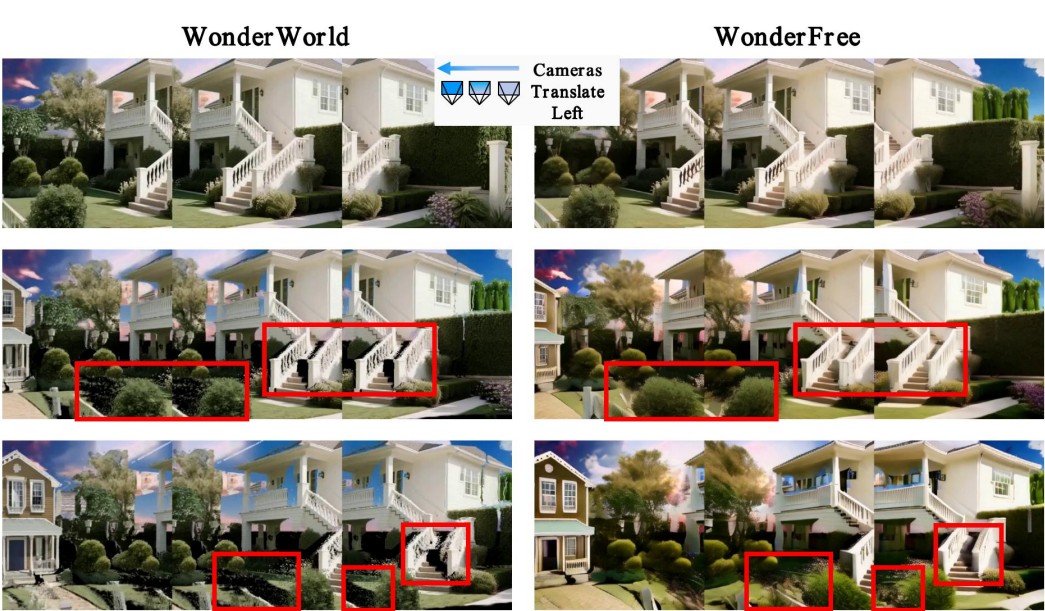

Figure 24: Qualitative examples.

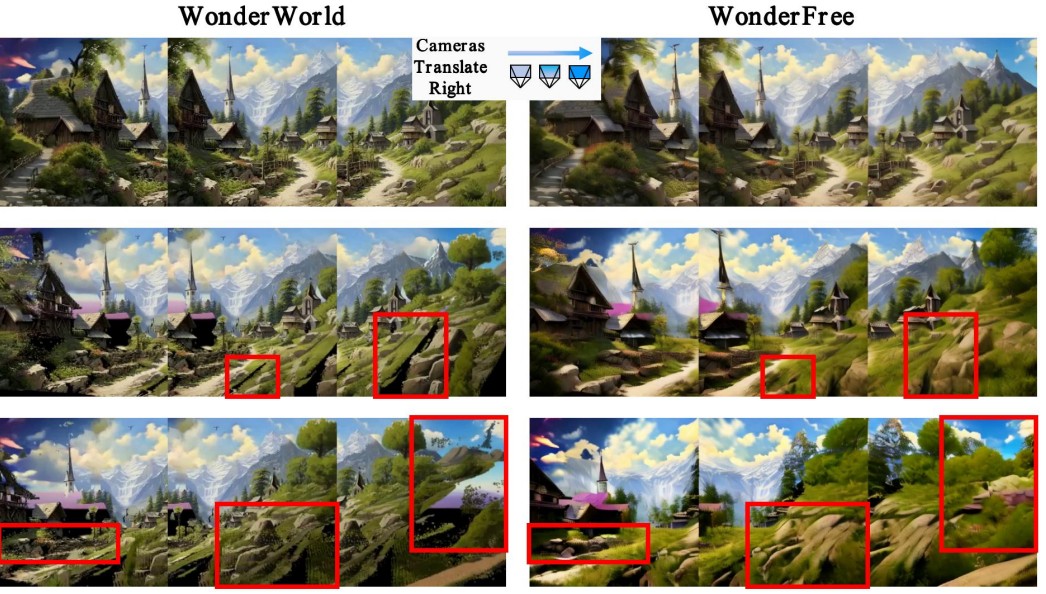

Figure 25: Qualitative examples.

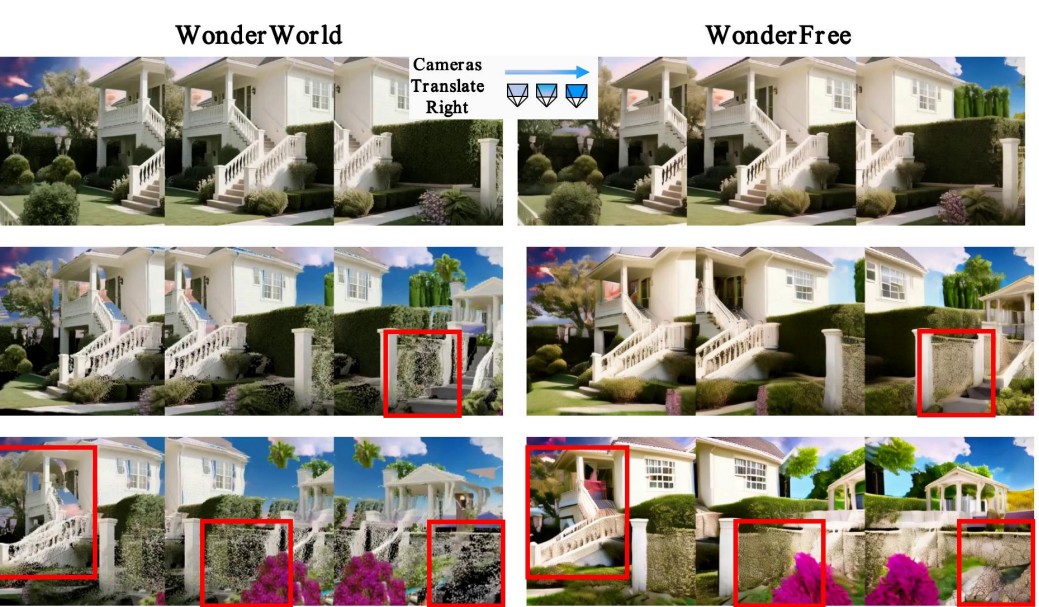

Figure 26: Qualitative examples.

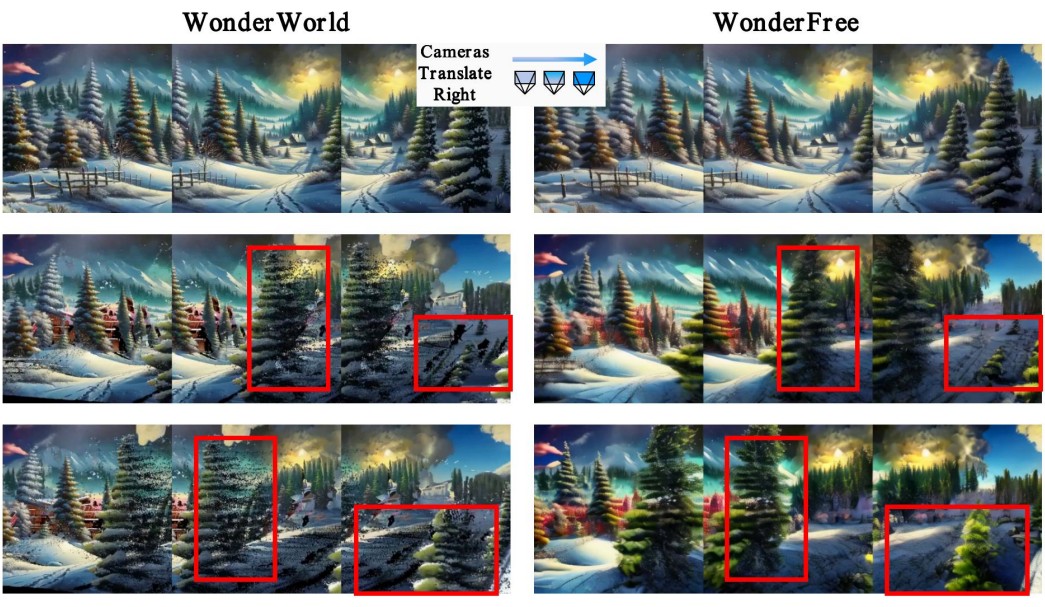

Figure 27: Qualitative examples.

