# OpenReview forum: "WonderFree: Enhancing 3D World Generation via Video Diffusion Prior with Multi-view Consistency"
_ICLR.cc/2026/Conference — ICLR 2026 Conference Withdrawn Submission_

### Official Review · Reviewer_1mvn · 2025-10-19

**Soundness:** 3
**Presentation:** 4
**Contribution:** 3
**Rating:** 4
**Confidence:** 5

**Summary:**

The paper proposes WonderFree, a 3D world generation framework that improves explorability from a single image by tackling two coupled challenges, novel‑view quality and cross‑view consistency, through an iterative render‑restore‑refine loop over a coarse 3D Gaussian world. The method introduces WorldRestorer, a diffusion‑based video restoration module trained on paired degraded versus clean videos produced via an under‑trained 3DGS pipeline, and ConsistView, a multi‑view joint restoration mechanism that concatenates overlapping viewpoint frames to enforce spatial coherence across views.

**Strengths:**

1. Clear problem decomposition into novel‑view quality and cross‑view consistency, mapping to dedicated modules that directly address observed failure modes in prior systems.​
2. WorldRestorer leverages video diffusion priors for restoration, effectively removing floaters and ghosting in novel‑trajectory renderings while preserving appearance cues.​
3. ConsistView introduces multi‑view joint restoration via overlapping viewpoint concatenation, improving spatial coherence across adjacent views in both training and inference.
4. WorldScopeDataset spans diverse scene styles (indoor, urban, natural, artistic) and multi‑view sequences, benefiting generalization and aesthetic quality.

**Weaknesses:**

1. Video Comparisons are limited. They only compare against WonderWorld and that too only on 4 scenes. Would be great if there could be more video comparisons on more scenes and other methods too like WonderTurbo etc.
2. No discussion of failure cases. When do these model fail? Is there a limit to the amount of camera movement? Ig so since in all video comparisons, I see only a limited camera motion and not beyond a limit. I urge the authors to discuss the limitations and future scope of the work.
3. How does this method work on real-world scenes? I see that the paper shows results on a lot of synthetic scenes in the appendix but I am unable to see any real-world results. How does the model perform when the initial image contains occlusions like say people, cars, animals etc. How does it treat the 3D of these occlusions? For example: when you move behind a person, does the model understand it as 3D or does it consider it as a 2D surface? More results on real-world results are needed.
4. In the video results and the image results, I see that Wonderfree tends to smoothen the result extremely compared to WonderWorld which is extremely bad. I am assuming it is because of the WorldRestorer producing averaged out results. Could the authors confirm the exact reason? For example: Fig 7, first row. The buildings in the background are sharper in WonderWorld but Wonderfree completely smoothens them out. Similarly in second row, the trees and the house gets smoothened out and in third row, the book shelf and the tree get smoothened out.
5. In Fig 11, the pillars on the staircase seems to be distorted in Wonderfree quite a bit whereas WonderWorld seems to do decently well. Even though there are a few artifacts which are not too noticable. As a user, I would definitely pick WonderWorld over wonderfree in this case. I see that this a recurring theme across various image results. WonderWorld tends to have a lot of graining artifacts where as WonderFree tends to smoothen the results or distort them. Since the WonderWorld contains grainy artifacts, I propose a simple baseline of WonderWorld + Difix3D which would remove artifacts and sharpen the results. I urge the authors to run this baseline and compare against Wonderfree. I am assuming WonderWorld + Difix3D would definitely be on-par or better than Wonderfree just due to the fact that Wonderfree tends to average/blur things out.
6. A more geometric metric like LPIPS would be a better way to quantify the difference. I feel that CLIP score are more like PSNR and wouldn’t be much affected by blurring, whereas metrics like LPIPS measure the geometric quality and would better quantify the geometry. Since GT are not available, LPIPS may not be possible to evaluate but it would be great if authors could find some non-ground truth based metrics like VBench/VBench-2 to quantify the geometric quality. Can you include geometry‑aware evaluations (e.g., depth consistency, reprojection error) or cross‑view photometric consistency to more directly quantify structural fidelity.
7. Even though WonderFree uses ConsistView to improve spatial consistency, I see that in Fig 1, the windows of the house are consistent in WonderWorld but get distored and are inconsistent in the case of WonderFree. Why is it the case? Can authors give a proper justification as to why this is the case?
8. How does the method extend to 360 degree cases? Does it model the geometry of the objects very well or does it distort it? Can authors provide a justification on this?

**Questions:**

I have highlighted the major questions in the weaknesses already, and I am summing them up below(I have summarised them shortly so that it is easier for the reviewer to quote the exact question they are answering. Please refer to the Weakness for the detailed problem and question asked.

1. Could you provide more extensive video comparisons on a larger set of scenes and include other relevant baselines such as WonderTurbo? Currently, the paper only compares against WonderWorld on four scenes, which feels too limited to properly judge performance.

2. When do these models fail? Is there a particular limit to how much the camera can move before the model starts breaking down? From the current video results, it seems that all camera motions are quite restricted. Could you discuss such limitations and describe future directions to overcome them?

3. How does the proposed method perform on real-world scenes? Most of the results in the appendix appear to be synthetic. It would be useful to see how the model handles real-world data where lighting, noise, and texture variation are more complex.

4. How does the model handle occlusions in the input image, such as people, cars, or animals? For instance, when the camera moves behind a person, does the model correctly understand it as a 3D object or does it simply treat it as a flat 2D surface? More real-world examples with occlusions would help clarify this.

5. In the qualitative comparisons, WonderFree seems to overly smooth out fine details compared to WonderWorld, which looks sharper but grainier. Is this smoothing effect caused by the WorldRestorer producing averaged results? Could the authors confirm the reason behind this behavior?

6. In Fig. 7, especially in the first three rows, why does WonderFree noticeably smooth out background details such as buildings, trees, and bookshelves, whereas WonderWorld preserves them more clearly? What causes this difference in texture sharpness between the two methods?

7. In Fig. 11, the pillars on the staircase appear significantly distorted in WonderFree while WonderWorld handles them better. This seems to be a recurring issue across multiple examples. Could you explain why WonderFree tends to introduce such geometric distortions?

8. Since WonderWorld produces grainy artifacts and WonderFree produces smooth or distorted results, could you try a simple baseline combining WonderWorld with Difix3D to reduce artifacts and improve sharpness? I suspect that such a combination might outperform WonderFree; could you test this baseline and include it in the comparisons?

9. For evaluation, can you include geometry-aware metrics such as LPIPS, depth consistency, reprojection error, or cross-view photometric consistency? CLIP scores alone seem insufficient since they don’t capture geometric or structural quality. If ground-truth data are unavailable, could you consider no-reference metrics like VBench or VBench-2 instead?

10. Even though WonderFree uses ConsistView to improve spatial consistency, I notice in Fig. 1 that the windows of the house become distorted and inconsistent compared to WonderWorld, which remains stable. Why does this happen, and can you provide a justification for this inconsistency despite the use of ConsistView?

11. How does your method generalize to 360-degree scenarios? Does it accurately model the 3D geometry of objects, or does it introduce distortions when the camera moves around the scene? Please provide some clarification or visual evidence supporting this.

---

### Official Review · Reviewer_P8eU · 2025-10-20

**Soundness:** 3
**Presentation:** 3
**Contribution:** 2
**Rating:** 4
**Confidence:** 4

**Summary:**

This paper proposes WonderFree, a framework to enhance 3D world generation from a single image by improving both novel view rendering quality and cross-view consistency.
The method introduces two key components:

1. WorldRestorer: a diffusion-based video restoration model trained to eliminate floaters, ghosting, and distortions in novel-view renderings produced by 3D Gaussian Splatting (3DGS).

2. ConsistView: a multi-view joint restoration mechanism that ensures spatial and temporal coherence across multiple viewpoints during video restoration.

To train WorldRestorer, the authors construct a new WorldScopeDataset, combining synthetic and real multi-view data with paired degraded and clean video sequences. The framework iteratively refines the coarse 3D world by rendering and restoring novel trajectories, progressively improving quality and consistency.

Experiments demonstrate that WonderFree outperforms baselines such as WonderWorld, DreamScene360, and LucidDreamer, achieving higher CLIP-based scores and a 77.2% user preference rate.

**Strengths:**

1. Originality: employ a video diffusion prior explicitly as a restoration module for 3D world refinement.

2. Quality: solid experimental setup, well-designed ablations, and both quantitative and qualitative validation.

3. Clarity: Excellent figures and detailed implementation explanation.

4. Significance: provides a clear path to improving 3DGS-based world generation, which is a key bottleneck in current methods.

**Weaknesses:**

1. The iterative rendering-restoration loop may be computationally intensive for long trajectories or high-resolution scenes.

2. The approach relies on 2D diffusion restoration; fully 3D-aware training (e.g., with volumetric priors) maybe could further improve spatial reasoning.

3. While the study isolates modules, it lacks detailed analysis of how different multi-view configurations (e.g., varying K, $\delta \Theta$) affect results.

**Questions:**

1. Could WorldRestorer be extended to operate directly in a 3D latent space rather than in the rendered 2D video domain?

2. The WorldScopeDataset covers multiple styles. How balanced are these subsets, and do style biases affect generalization?

---

### Official Review · Reviewer_LFTP · 2025-10-27

**Soundness:** 2
**Presentation:** 3
**Contribution:** 2
**Rating:** 4
**Confidence:** 4

**Summary:**

This paper presents WonderFree, which is developed to enhance the visual quality and consistency of 3D worlds generated by WonderWorld. In particular, it includes a diffusion-based video restoration model and to improve quality of rendered views, and it also integrates a ConsistView mechanism to enhance consistency across multiple views. The author also built a large-scale WorldScopeDataset to train the restoration network.

**Strengths:**

The paper is well written, and the proposed pipeline is logically reasonable as an extension of WonderWorld. The implementation is solid, and the experimental results have shown improvements and been well presented.

The contribution of the WorldScopeDataset (23 M images, 6 K+ scenes) that combines real and synthetic multi-view video pairs is also valuable.

**Weaknesses:**

The overall contribution seems incremental. This paper is mainly to add an iterative restoration and fine-tuning stage on top of existing 3D reconstruction framework, i.e., WonderWorld, without fundamentally improving the core 3D generation mechanism.

The proposed distillation scheme that leverages diffusion or image priors for restoration has been explored in many prior works, and its novelty is limited.

The framework depends on both the quality of the initial 3D reconstruction and the performance of the diffusion model in correcting errors. This kind of dual dependency probably makes the restoration quality unstable and potentially introduces artifacts or degradation into the 3D model itself.

The complexity and limited scalability. The iterative multi-view video restoration and 3D refinement loop might be computationally expensive, so it's hard to generalize or deploy at scale.

**Questions:**

In additional to the weakness part, the authors are suggested to discuss analyze the computational complexity and efficiency.

Is WorldRestorer designed only for refining WonderWorld-generated scenes, or can it generalize to other 3D reconstruction systems?

---

### Official Review · Reviewer_6oJq · 2025-10-31

**Soundness:** 3
**Presentation:** 2
**Contribution:** 2
**Rating:** 2
**Confidence:** 4

**Summary:**

1. The paper introduces WonderFree, a new system for generating less-artifacts explorable 3D worlds from a single image.
2. WonderFree introduce two components: WorldRestorer, which is a finetuned video diffusion models to clean up visual artifacts in new views, and ConsistView, which is a technique that processes multiple camera views at once to make sure the entire world remains spatially consistent.
3. The paper introduces a dataset called WorldScopeDataset that can gather training data for WorldRestorer.

**Strengths:**

1. The paper propose a novel WorldRestorer component that significantly cleans up artifacts in the pre-built coarse 3D world, a key technical contribution visually supported by the visualization in paper and supplementary material.
2. The overall WonderFree framework effectively achieves its goal of generating a more cohesive and explorable 3D world from a single input image.
3. A new dataset, WorldScopeDataset, is introduced to facilitate the training of the restoration model. The dataset contains paired videos which are ground truth videos and matched degraded rendered videos.
4. The paper includes comprehensive experiments and a positive user study, which convincingly validate the performance improvements over prior state-of-the-art methods.

**Weaknesses:**

1. The proposed WorldRestorer appears to be an incremental contribution built primarily on the previous WonderWorld model, which limits its novelty. It closely resembles existing works like Reconfusion and Difix3D+. Crucially, its utility seems restricted to WonderWorld's pre-built coarse worlds, and its generality for other Gaussian-Splatting based degraded videos needed to be clarified.
2. The presented results are incomplete. The paper should feature full panorama results of the reconstructed world, not just a limited set of viewpoints compared against WonderWorld, to properly validate the global coherence claim.
3. Efficiency is another concern. The introduction of video diffusion models likely adds significant latency. A critical analysis of the framework's runtime and throughput is necessary for acceptance.
4. The overall quality remains limited. The supplementary video (comparison1.mp4) suggests that the restored scenes suffer from blurriness and a loss of fine detail when compared directly to the WonderWorld output. This may indicate inefficient training of video diffusion model.

**Questions:**

1. How many novel camera paths are required in a typical exploration session to achieve high quality? Furthermore, does the system process this refinement in parallel or sequentially, which has massive implications for latency?
2. The evaluation relies solely on a CLIP-based score, which only measures visual-text consistency. Why do the authors not provide metrics that evaluate the visual fidelity of the generated content itself as some blurs are seen in the demo video?
3. What is the underlying video diffusion model architecture used as the base for the WorldRestorer?

---

### Note · Authors · 2025-11-13

I have read and agree with the venue's withdrawal policy on behalf of myself and my co-authors.